# Robust Point Cloud Registration for Aircraft Engine Pipeline Systems

**DOI:** 10.3390/s24113358

**Published:** 2024-05-24

**Authors:** Yusong Liu, Zhihai Wang, Jichuan Huang, Liyan Zhang

**Affiliations:** 1College of Mechanical and Electrical Engineering, Nanjing University of Aeronautics and Astronautics, Nanjing 210016, China; liuys007@avic.com; 2Chengdu Aircraft Industrial (Group) Co., Ltd., Chengdu 610091, China; jichuan1980@163.com; 3Norla Institute of Technical Physics, Chengdu 610041, China; w13518108180@163.com; 4Systems Engineering, Northwestern Polytechnical University, Xi’an 710072, China

**Keywords:** point cloud registration, aircraft engine pipeline system, feature descriptor

## Abstract

Aircraft engine systems are composed of numerous pipelines. It is crucial to regularly inspect these pipelines to detect any damages or failures that could potentially lead to serious accidents. The inspection process typically involves capturing complete 3D point clouds of the pipelines using 3D scanning techniques from multiple viewpoints. To obtain a complete and accurate representation of the aircraft pipeline system, it is necessary to register and align the individual point clouds acquired from different views. However, the structures of aircraft pipelines often appear similar from different viewpoints, and the scanning process is prone to occlusions, resulting in incomplete point cloud data. The occlusions pose a challenge for existing registration methods, as they can lead to missing or wrong correspondences. To this end, we present a novel registration framework specifically designed for aircraft pipeline scenes. The proposed framework consists of two main steps. First, we extract the point feature structure of the pipeline axis by leveraging the cylindrical characteristics observed between adjacent blocks. Then, we design a new 3D descriptor called PL-PPFs (Point Line–Point Pair Features), which combines information from both the pipeline features and the engine assembly line features within the aircraft pipeline point cloud. By incorporating these relevant features, our descriptor enables accurate identification of the structure of the engine’s piping system. Experimental results demonstrate the effectiveness of our approach on aircraft engine pipeline point cloud data.

## 1. Introduction

The pipeline system in aircraft engines is a critical component responsible for functions such as fuel delivery, exhaust gas circulation, and signal transmission. It consists of numerous structures, including pipes, rings, and brackets, that must be assembled correctly to ensure optimal engine performance and safety. Regular inspections are necessary to ensure the correctness of pipeline assembly, including the positioning of pipelines and the accuracy of pipeline connections. However, traditional inspection methods, such as visual assessment or manual measurements, are limited in their ability to accurately and efficiently examine complex pipeline systems.

In recent years, point-cloud-based 3D detection techniques have become increasingly prevalent in the field of aircraft engine and airplane manufacturing. These techniques have been applied to various applications, such as measuring gaps between aircraft engine pipelines [1], assessing seam measurements on aircraft surfaces [2], detecting surface deformations [3,4], and identifying rivets [5,6]. By leveraging 3D point cloud data, these techniques enable highly precise and quantitative measurements of object geometries. The detailed representation provided by point clouds allows for accurate and reliable analysis of various components in aircraft manufacturing. Moreover, the non-contact nature of 3D scanners offers a distinct advantage in quality inspection, as it eliminates the need for physical contact with delicate aircraft structures. Given these advantages, employing point-cloud-based methods for inspecting aircraft engine pipeline systems is a natural choice. These methods address the limitations of manual inspection techniques and provide a more efficient and accurate approach to assessing the integrity and quality of the pipeline system.

Traditional techniques involve laser tracking or using reference markers to match point clouds captured from different angles. However, for aeroengines, the closed environment and high-temperature, high-pressure working conditions pose challenges to these methods. The laser tracking equipment has a blocked field of vision in a closed and narrow environment. Global tracking of the work environment is difficult, and the high-precision production environment of aircraft engines does not allow for additional reference markers. Therefore, we need to complete the measurement data through the registration and fusion of point clouds obtained from multiple viewpoints. Therefore, automatic registration of point clouds from different viewpoints through registration algorithms becomes a critical prerequisite for aircraft engine pipeline system inspection.

As depicted in Figure 1, aircraft engine pipeline systems are characterized by complex structures and intricate layouts, leading to substantial internal occlusions. Consequently, the acquired point cloud data of most pipelines often suffer from missing information. Furthermore, cylindrical pipelines exhibit similar structures when observed from different viewpoints, lacking distinctive registration features. These characteristics pose challenges for existing 3D registration methods, such as iterative closest point (ICP) [7] and normal distributions transform (NDT) [8], in achieving high-precision registration.

In this paper, we propose a novel registration method that addresses these challenges by incorporating two key ideas. Firstly, we introduce a new descriptor called point line–point pair features (PL-PPFs) [9], which leverages the point line features of aircraft pipeline systems. PL-PPFs effectively represent the structural characteristics of the pipelines, even in the presence of incomplete point cloud data. Secondly, we present an effective evaluation strategy to eliminate ambiguities in PL-PPFs. This allows us to establish correct correspondences between point cloud pairs and achieve accurate registration. Lastly, we establish a hash table to assess the similarity of PL-PPF descriptors, obtain correspondence relationships, and solve the transformation matrix. Experimental results demonstrate that our method achieves a higher registration accuracy compared to other state-of-the-art methods.

In summary, our contributions are three-fold:We present a fully automated registration framework that enables the acquisitionof accurate and complete point clouds of aircraft engine pipeline systems forinspection purposes.We introduce a novel 3D point pair feature descriptor, called PL-PPFs, which effectively captures the characteristics of pipelines and engine assembly features in the aircraft pipeline system.We propose an effective evaluation strategy, referred to as ambiguity elimination, to select the most suitable combinations of point and line features in PL-PPFs, reducing ambiguities and improving the accuracy of the registration process.

## 2. Related Work

Point cloud registration is a hot research topic in various fields, including computer vision, shape recognition, and robotics [10,11]. Point cloud registration methods can be broadly categorized into optimization-based methods, feature-based point matching, and end-to-end learning. In this section, we provide a review of these types.

### 2.1. Optimization-Based Methods

The random sample consensus (RANSAC) algorithm [12] is one of the earliest methods used to perform registration between two images based on a parameterized geometric relationship, such as a projective transformation or epipolar geometry. The RANSAC algorithm can optimize and accelerate point cloud registration from a computational perspective. It follows a hypothesis and verification strategy: it repeatedly samples a minimal subset of data (e.g., four correspondences for a projective transformation or seven for a fundamental transformation), estimates a model as the hypothesis, and verifies its quality based on the number of consistent inliers. The final correspondences consistent with the optimal model are identified as inliers. However, RANSAC has some limitations, including sensitivity to parameter selection, high computational complexity, low efficiency, and sensitivity to noise and outliers.

In recent years, various improvements have been proposed to address the sensitivity of RANSAC to noise and outliers in point cloud registration. For instance, the MLESAC algorithm [13] is used for model estimation and outlier removal, providing more robust model estimation by considering noise and outliers in the data. LO-RANSAC (locally optimized RANSAC) [14] is a RANSAC-based point cloud registration algorithm aimed at improving the robustness and accuracy of registration. It achieves more accurate results by locally optimizing the estimated model parameters at each RANSAC iteration. PROSAC (progressive sample consensus) [15] reduces the computational complexity and the need for extensive trial computations in RANSAC by introducing a priority mechanism and dynamic sampling strategy, resulting in more efficient and robust model estimation. NAPSAC [16] addresses the performance degradation of traditional RANSAC in high-dimensional space by gradually improving the accuracy of model parameters and the quality of inliers through step-wise optimization and selection. This allows for better handling of noise and outliers in high-dimensional space, leading to more reliable model estimation and more accurate and robust point cloud registration results.

### 2.2. Feature-Based Point Matching

Selecting appropriate feature descriptors is a critical step in achieving successful feature matching during point cloud registration. It involves extracting point feature descriptors by analyzing local and possibly global 3D geometry rather than relying solely on point coordinates to establish correspondences. Classic methods often employ handcrafted feature descriptors, which can be broadly categorized as spatial distribution histogram-based and geometric attribute histogram-based descriptors [16]. The spatial distribution histogram-based descriptors represent local features by generating histograms from statistical data of geometric attributes, such as normals and curvatures, within a support region. The distribution of these attributes is analyzed to capture the local geometric characteristics of the point cloud. On the other hand, geometric attribute histogram-based descriptors construct a local reference coordinate system for each keypoint in the point cloud. This divides the 3D region surrounding the keypoint into multiple boxes, forming a histogram where the value of each box is determined by accumulating spatial distribution measurements. These descriptors take into account the geometric attributes in a local region to characterize the point cloud’s features. Several common feature descriptors include point feature histograms (PFHs) [9], fast point feature histograms (FPFHs) [17], Signature of Histograms of OrienTations (SHOT) [18], and point feature histograms with RGB (PFHRGBs) [19]. These descriptors play a crucial role in feature matching and are widely used in various point cloud registration tasks. However, in the face of complex and mutually occluding pipelines, it is difficult to use these descriptors in point cloud feature extraction and registration. Therefore, a novel feature descriptor is urgently needed to avoid incomplete point cloud registrations.

### 2.3. End-to-End Learning

Using convolutional neural networks (CNNs) on sparse point cloud data poses challenges due to their unordered and scattered nature. However, the adoption of end-to-end networks for point-based tasks has been gaining attention, especially in the context of registration problems. End-to-end networks facilitate the establishment of correspondences between points in two point clouds, the filtering out of unreliable correspondences, and the estimation of transformations based on optimal point pairs.

PointNetVLAD (PointNet with VLAD) [20] employs the PointNet network architecture to extract local features from point clouds and aggregates these features into a global descriptor using the VLAD (vector of locally aggregated descriptors) encoder. It achieves registration by learning to register point clouds to the global descriptor. DeepGlobalRegistration [21] learns the global transformation between point clouds by minimizing a loss function to optimize the registration of point clouds. Deep closest point [22] learns to register two point clouds to a shared reference point cloud, achieving registration. DCP utilizes a PointNet encoder to extract features and employs nearest neighbor search and differentiable optimization methods to optimize the registration. DCP is efficient, robust, and capable of handling non-rigid transformations. PointNetLK [23] combines the ideas of nearest neighbor search and the iterative closest point (ICP) algorithm. PRNet employs a PointNet encoder to extract features from point clouds and iteratively improves the registration of point clouds through optimization. It gradually refines rigid transformations through multiple iterations to achieve precise point cloud registration. RPM-Net [24] utilizes a PointNet encoder to extract features from point clouds and combines the RPM (randomized point matching) framework for point cloud matching and registration. RPM-Net matches point clouds by learning a similarity measure between features and utilizes differentiable optimization methods for registration. More recently, Chen et al. proposed ImloveNet [25], which employs a misaligned image to support low-overlap point cloud registration.

Indeed, employing deep learning for point cloud registration necessitates substantial data for effective learning. In the specific context of aircraft detection, creating large-scale datasets is challenging, and publicly available datasets might not adequately cover the diversity of pipeline systems.

## 3. Hardware System

As shown in Figure 2, our hardware setup consists of a structured light 3D camera, a workstation, and a power supply. To meet the requirements of aircraft engine pipeline detection, we have chosen the ZHIXIANG Optoelectronics Surface HD 100 as our 3D camera. This camera utilizes infrared laser technology as the light source and has an optimal working distance of 900 ± 500 mm, with a maximum working distance of 1400 mm. It provides repeatability of ±0.25 mm. The camera dimensions are 240 × 70 × 43.6 mm, and it offers a depth map resolution of 1920 × 1200 and a color image resolution of 1920 × 1080.

In our hardware setup, the 3D scanner is mounted on a support structure consisting of a tripod and a gimbal. The 3D scanner is connected to the workstation and the power supply using two cables. During operation, the 3D scanner captures point clouds of the aircraft engine pipeline system. These point clouds are then transmitted to the workstation for registration and analysis. After each data acquisition is completed, the tripod is manually repositioned to the next station until the entire surface of the pipeline system is covered by the 3D scanner. It is important to note that the raw scanned point cloud data obtained from the 3D scanner is in the scanner’s coordinate system. Therefore, we employ our proposed method to automatically transform the data into a global coordinate system.

## 4. Method

### 4.1. Overview

The method takes the raw scan data obtained from the 3D scanning laser, which contain the pipeline features, as input. Our goal is to register the raw scan pipeline point clouds, consisting of repetitive and incomplete structures, into a global point cloud. Figure 3 provides an overview of the proposed framework, which consists of feature extraction, feature descriptor construction, and similarity matching.

**Feature Extraction.** Firstly, we extract both line features and pipeline features from the raw scan data containing the aircraft engine pipeline system. The line segments are detected from the raw scan data, and any noisy line segments that might compromise the robustness of the feature descriptors are filtered out from the extracted results. For pipeline feature extraction, we utilize a specially designed scanning probe to detect the pipeline structure, and then obtain the pipeline features from the detected pipeline structure slices. These pipeline features are represented by axial feature points, which are obtained by fitting discrete points on the pipeline centerline from local patches of the point cloud.

**Descriptor Design.** With the extracted line segment features and pipeline features, we proceed to construct the PL-PPF descriptor. To ensure the robustness of the descriptor, we propose an effective judgment strategy to determine the combinations of line segments and pipeline feature points that are suitable for constructing the PL-PPF descriptor. This strategy helps us in selecting the most appropriate combinations to accurately represent the geometric relationships between the line segments and pipeline features.

**Correspondence Determination.** Once the PL-PPF descriptor is constructed based on the selected combinations, we use it to establish correspondences between two point clouds, enabling accurate registration of the raw scan point pairs. We achieve this by storing the PL-PPF descriptors in a hash table and then querying the hash table with key values to identify similar descriptors between the two point clouds. This allows us to determine all potential correspondences between the point clouds. To improve the efficiency of the algorithm and eliminate any ambiguous correspondences, we further filter out error correspondences through constraint conditions. This step ensures that only reliable correspondences are considered in the registration process. Finally, with the filtered correspondences, we solve for the matching matrix using singular value decomposition (SVD), which yields the final transformation for matching the raw scan point clouds. The proposed registration method, based on the PL-PPF descriptor and the subsequent correspondences, achieves accurate and complete point cloud registration of the aircraft engine pipeline system.

### 4.2. Feature Extraction

We extract both line features and pipeline point features from the raw scan data to construct new point pair features.

To extract line features, we utilize a 3D line segment detection method [26]. However, the extracted line segments may include discontinuous lines or non-engine surface contour segments at pipeline joints, which are not suitable for robust feature descriptor construction. To address this issue, we divide the line segments into smaller segments, ensuring that the segments extracted from both point clouds have comparable lengths. This approach improves the accuracy of matching and registration. This ensures that we focus on relevant and meaningful features for constructing our feature descriptors.

To extract pipeline features, we introduce a novel point cloud pipeline detection method, which enables us to segment the pipeline part and subsequently extract the feature points of the segmented pipeline. Segmenting the pipeline is a crucial step for facilitating accurate feature extraction. Traditional segmentation methods often struggle with the complexity of the pipeline system, which includes numerous pipeline joints and background information. Consequently, they fail to provide precise segmentation of the pipeline point cloud. Additionally, occlusions lead to incomplete point cloud data, and the majority of aircraft pipeline systems consist of curved pipes, making cylinder fitting ineffective for extracting pipeline information.

Motivated by these challenges, we first design a detection approach to identifying pipeline structures. As shown in Figure 4, we randomly select a point, denoted as $p_{m}$. To determine whether $p_{m}$ is a point on the pipeline, we define a spherical neighborhood slice, $S_{m}$, around $p_{m}$ with a radius *r*. The value of *r* is determined based on empirical knowledge of the pipeline radius range in the aircraft pipeline system. We then project the normals, $N_{m}$, of the points in $S_{m}$ onto a Gaussian sphere and perform plane fitting on the projected points on the Gaussian sphere. Let $n_{i}\in N_{m}$ and $n_{i}=\left(x_{ni},y_{ni},z_{ni}\right)$. Suppose that we fit it by ax+by+cz−1=0, i.e., $M^{T}A=L_{1}$, where
(1)$$M={\left(\begin{array}{cccc} x_{n1} & x_{n2} & \cdots & x_{nM}\\ y_{n1} & y_{n2} & \cdots & y_{nM}\\ z_{n1} & z_{n2} & \cdots & z_{nM} \end{array}\right)}^{T}$$
(2)$$A={\left(\begin{array}{ccc} a & b & c \end{array}\right)}^{T}$$
(3)$$L_{1}={\left(\begin{array}{cccc} 1 & 1 & \cdots & 1 \end{array}\right)}^{T}$$

After fitting the plane, its normal vector $v_{m}$ represents the axis direction of slice $S_{m}$. Using $v_{m}$ and $p_{m}$, we can determine a plane denoted as α. We then project the points of $S_{m}$ onto the plane $\alpha_{m}$, resulting in new points $T_{m}$. For each point $s_{i}\in S_{m}$ and its corresponding point $t_{i}\in T_{m}$, we perform circle fitting based on $v_{m}$. The fitted circle center $O_{m}$ is considered as a discrete point on the pipeline axis at $p_{m}$, and the radius $R_{m}$ is considered as the pipeline radius at that point.

Next, we randomly select *N* points from the slice $S_{m}$ and denote them as $q_{i}$. For each $q_{i}$, we repeat the above steps, considering it as the center of a sphere with radius *r* and segmenting its spherical neighborhood slice. Circle fitting is performed on the segmented slice, resulting in the fitted circle radius $R_{i}$ for each $q_{i}$. In our experiments, we set *N* to 7 to evaluate whether $p_{m}$ is a point on the pipeline.

For $p_{m}$, we obtain N + 1 slices and their corresponding fitted circle radii. We calculate the average radius μ and the standard deviation of these radii. We define thresholds $\mu_{0}$ and λ as the criteria for determining seed points. If $\mu_{0}-\lambda \leq \mu \leq \mu_{0}+\lambda$, we consider $p_{m}$ as a seed point for extracting pipeline features, and $o_{m}$ is the extracted pipeline feature point. If μ is not within the threshold range, we consider $p_{m}$ as a non-seed point for extracting pipeline features. Regardless of whether $p_{m}$ is a seed point or not, once the analysis of $p_{m}$ is completed, it, along with the slices $S_{m}$ generated by segmenting the point cloud, is marked as used in the original point cloud. Then, we randomly select another point from the remaining unused points and repeat the above operations until all points in the original point cloud are marked as used.

### 4.3. Descriptor Design

The point pair feature (PPF), originally proposed by Drost, B. et al. [9], is a well-established and effective local descriptor widely used in point cloud registration. It involves a pair of points, $p_{1}$ and $p_{2}$, along with their associated normal vectors, $n_{1}$ and $n_{2}$. The PPF descriptor, denoted as $D(p_{1},p_{2})$, is formulated as follows:(4)$$D\left(p_{1},p_{2}\right)=\left(∥d∥,∠\left(n_{1},d\right),∠\left(n_{2},d\right),∠\left(n_{1},n_{2}\right)\right)$$
where *d* is computed as the difference between the coordinates of $p_{1}$ and $p_{2}$, i.e., $d=p_{1}-p_{2}$. The term $\theta (n_{1},d)$ indicates the angle between the normal vector $n_{1}$ and *d*, with the angle range constrained within [0,π]. Similarly, $\theta (n_{2},d)$ captures the angle between $n_{2}$ and *d*, while $\theta (n_{1},n_{2})$ characterizes the angle between the two normal vectors $n_{1}$ and $n_{2}$.

We have developed a new PPF (point pair feature) called a PL-PPF, based on the contour line features of the engine assembly and the point features of the pipeline axis. The construction process of the PL-PPF includes the following steps.

To represent the line features, we use the midpoints, denoted as $P_{Li}$, obtained from the smaller line segments. Each midpoint $P_{Li}$ captures the direction of its corresponding line segment. Therefore, we define sets of line segments as $L_{i}$, represented by $L_{i}\left(P_{Lij},n_{Li}\right)$, where $P_{Li}$ represents the midpoint of the line segment $L_{i}$ and $n_{Li}$ represents the direction of $L_{i}$. Similarly, we denote the corresponding midpoints as $P_{l}$, where $p_{L}\left(x_{j},y_{j},z_{j}\right)$ represents the midpoint of the line segment *L* and $n_{L}$ represents its direction.

**Validity judgment.** We propose an effective validation strategy to filter the extracted line features. The PL-PPF constructed directly from line segment $L_{i}$ and vertex $P_{Vi}$ may be ambiguous in many cases, as shown in Figure 5. To improve the robustness of registration, we validate the effectiveness of the combinations between line segments and vertices before constructing the PL-PPF. To achieve this, we have introduced a scoring function called PLScore that evaluates the effectiveness of the extracted line segments in constructing unique PL-PPFs. Line segments may have different PLScores depending on their positions in the plane. We select line segments with higher scores as valid features because unique PL-PPFs can be constructed using high-scoring line segments. More specifically, the PLScore can be represented as follows: $$\begin{array}{cc} \mathrm{PLScore}=\frac{1}{3}[ & \sigma_{1}\left(d_{1},d_{2},d_{3},\ldots ,d_{i}\right)\\ \; & +\sigma_{2}\left(d_{1}^{\prime },d_{2}^{\prime },d_{3}^{\prime },\ldots ,d_{i}^{\prime }\right)\\ \; & +\sigma_{3}\left(d_{1}^{\prime \prime },d_{2}^{\prime \prime },d_{3}^{\prime \prime },\ldots ,d_{i}^{\prime \prime }\right)] \end{array}$$
where
(5)$$\sigma =\sqrt{\frac{1}{N}\sum_{i=1}^{N}{\left(d_{i}-\mu \right)}^{2}}$$

In the equation, μ represents the average value of $d_{i}$, where *d* is the distance from the rivet vertex to each point on the line segment, and $d_{1}^{\prime }$, $d_{2}^{\prime }$, and $d_{3}^{\prime }$ represent the distances from $m_{1}$, $m_{2}$, and $m_{3}$, respectively. The subscript *i* of $d_{i}$ depends on the number of pipeline axis feature points.

In Figure 6, we demonstrate the process of selecting the starting point $m_{1}$, endpoint $m_{3}$, midpoint $m_{2}$, and line segment $L_{1}$ from the extracted line segments. We then calculate the distances from each selected point to all rivet vertices. As a result, each line segment is represented by three points, and this process is repeated for all extracted line segments. The obtained scores for all line segments represent their contributions to the construction of the PL-PPF. Afterward, we retain the line segments with relatively high scores. The retained line segments, denoted as $L_{vi}$, are used to construct the PL-PPF along with their corresponding vertices $P_{vi}$. We represent the set of effective line segments and their corresponding vertices as $G_{i}\left(P_{Vi},L_{Vi}\right)$. Our method can adaptively select effective line segments and vertices for different types of aircraft pipeline data, ensuring the uniqueness of the PL-PPF.

**PL-PPF Construction.** We construct a two-dimensional feature descriptor called PL-PPFs based on the line features derived from the tank contour and the point features from the pipeline axis. The features consist of angle and distance. In this descriptor, a directed point $p_{L}$ belongs to the selected line segment $L_{v}$, while the other common point $p_{V}$ comes from the corresponding pipeline feature point $p_{V}$ in the effective group *G*. Specifically, we define the feature descriptor *D* as follows:(6)$$\left(p_{L},p_{V}\right)=\left(∥d∥,∠\left(n_{L},d\right)\right)$$
where $d=p_{L}-p_{V}$. Once the local descriptor is given, we can construct the PL-PPF of the aircraft tank using the line segments and vertices. Given a set of valid lines and vertices $G_{i}\left(P_{Vi},L_{Vi}\right)$ in a point cloud frame, we construct the PL-PPF $F=\left\{D_{i}\right\}$ by calculating the equation. Note that a certain midpoint $p_{Li}$ in $L_{vi}$ needs to be combined with each vertex $p_{\mathrm{vi}}\in P_{v_{i}}$ to compute the PL-PPF $D_{i}$. Then, we compute the PL-PPFs for all other midpoints in the line segment $L_{vi}$. Finally, we repeat the above process for all elements in group $G_{i}$. This completes the construction of the local descriptor PL-PPFs for a point cloud frame.

### 4.4. Correspondence Determination

Unlike the common registration process that matches two scenes by iteratively searching for correspondences, our method first obtains corresponding points from similar feature descriptors and then filters the correspondences using constraints. Finally, we compute the transformation matrix using the filtered correspondences without iteration. The specific procedure is as follows:

**Hash table creation.** In this section, we construct a hash table for each point cloud to improve the accuracy and search speed of evaluating descriptor similarity. The PL-PPF is a 2D vector composed of distance and angle, which are different physical quantities with different scales. Therefore, we cannot measure them in the same way. To accurately assess the feature descriptor, we sample the distance and angle separately with different step sizes [9] and then merge the results into a vector, which becomes the key value $K_{i}$ for the subsequent hash table. The hash table created for a point cloud frame consists of three parts: the key value, PL-PPF, and corresponding coordinates. Specifically, we first extract valid groups $G_{i}$ of lines and vertices from a point cloud data frame. Then, for a pair of points $\left(p_{vi},p_{Li}\right)$ in group $G_{i}$, we construct a PL-PPF $D_{i}$ and calculate the key value $d_{i}$. In this way, we obtain an element of the hash table, including a key value, a PL-PPF $D_{i}$, and the corresponding coordinates $\left(p_{vi},p_{Li}\right)$. Finally, by repeating this process, we establish a complete hash table *T* for a frame of data.

**Evaluation of PL-PPF Similarity.** Given a pair of point clouds, *P* and *Q*, and their corresponding hash tables, we can establish the correspondence between the two point clouds by finding similar PL-PPF descriptors within the hash tables. First, we construct a hash table, $T_{\mathrm{tgt}\;}$, for the target point cloud. Additionally, we compute the PL-PPF descriptor, $D_{pi}$, from the valid group, $G_{i}$, of the source point cloud, and further calculate its key value, $K_{i}$. Using the key value, $K_{i}$, we can locate the corresponding position in the hash table, $T_{\mathrm{tgt}\;}$, and consider the PL-PPF descriptor, $D_{qi}$, with the same key value as $D_{pi}$. Similarly, we save the coordinates of the corresponding point, $q_{i}\left(q_{vi},q_{Li}\right)$, from the PL-PPF descriptor, $D_{qi}$, in $T_{\mathrm{tgt}\;}$, and also store the coordinates of the point, $p_{i}\left(p_{vi},p_{Li}\right)$, from $D_{pi}$. We represent this correspondence as $\mathrm{Cor}\left(p_{i},q_{i}\right)$. Finally, by repeating this process for all elements of $D_{pi}$ in the source point cloud, we obtain the initial correspondence set, $\mathrm{Cor}\left(p_{i},q_{i}\right)$, between the adjacent point clouds.

**Calculation of the Registration Matrix**. Given the correspondence relationship $\mathrm{Cor}\left(p_{i},q_{i}\right)$ between two point sets, *P* and *Q*, we need to compute a rigid transformation matrix, T, to register the source point cloud, *P*, with the target point cloud, *Q*. First, we establish an objective function defined by the distances between corresponding points in $\mathrm{Cor}\left(p_{i},q_{i}\right)$. By minimizing this objective function, we can obtain the transformation matrix, *T*, that registers the source point set, *P*, to the target point set, *Q*. The transformation matrix, *T*, is a 4x4 matrix composed of a rotation matrix, *R*, and a translation vector, *t*. Specifically, the objective function can be defined as follows:(7)$$E\left(T\right)=\sum_{i}{\left(Tp_{i},q_{i}\right)}^{2}=\sum_{i}{\left[\left(Rp_{i}+t\right)-q_{i}\right]}^{2}$$

To obtain the optimal transformation matrix *T*, we solve this problem using SVD, similar to the ICP (iterative closest point) process [7].

## 5. Experimental Results and Discussion

In this section, we present the experimental results of our pairwise registration method on point clouds acquired from various aircraft engine ducting systems. We aim to demonstrate the superiority of our method by comparing it with other registration techniques, specifically on engine gas ducts and oil ducts. The raw scan data used in our experiments are obtained through a structured light 3D scanner, and we use a laser tracker to serve as the ground truth for accurate registration. It is important to highlight that our experimental setup for aircraft engines allows us to obtain ground truth data using the combination of the structured light 3D scanner and laser tracker. However, we acknowledge that, in real manufacturing scenarios, the aircraft fuel tank is enclosed, with only a few small openings accessible, making it impractical to use a laser tracker in such manufacturing environments.

We conducted tests on four sets of aircraft engine gas ducting system data and five sets of aircraft engine oil ducts to evaluate the performance of our method. The registration results for the gas ducts are presented in Figure 7, while the results for the oil ducts are shown in Figure 8. Additionally, we provide quantitative comparisons of the registration results in Table 1 and Table 2, where we assess the registration error using the root mean square error (RMSE) between the transformed point clouds and the ground truth.

### 5.1. Gas Ducts of Aircraft Engines

The gas pipeline system in an aircraft engine is a crucial and complex component responsible for guiding air and fuel into the engine to facilitate combustion and generate thrust. This system consists of multiple pipes, valves, and fittings that ensure the proper flow of air and fuel according to design requirements, meeting the engine’s operational needs. Components like air conditioning pipes, vent pipes, and bleed pipes are part of the gas system, featuring larger pipe diameters and requiring additional support and fixation.

Figure 7 showcases the results of two sets of original scanned aircraft engine gas pipeline system data using various registration methods. Our algorithm demonstrates superior performance compared to other registration techniques. It is evident that the results of each algorithm are inferior to ours. Particularly, FGR and RANSAC + PPF lead to misregistration of the pipeline system, causing incorrect alignment of the pipes within the system. However, our method accurately matches them with the ground truth. The reason for this lies in the similarity of pipeline features within the aircraft system and the presence of numerous occlusions in the point cloud. These factors make conventional coarse registration techniques unable to precisely distinguish between different features. However, our method can correctly identify the pipeline structure and the engine’s outer wall assembly features by leveraging the geometric relationship between the pipeline’s axis feature points and the outer wall’s line segments to establish reliable correspondences. Consequently, our method successfully achieves accurate point cloud registration of the original scanned aircraft engine gas pipeline system.

**Figure 7 sensors-24-03358-f007:**
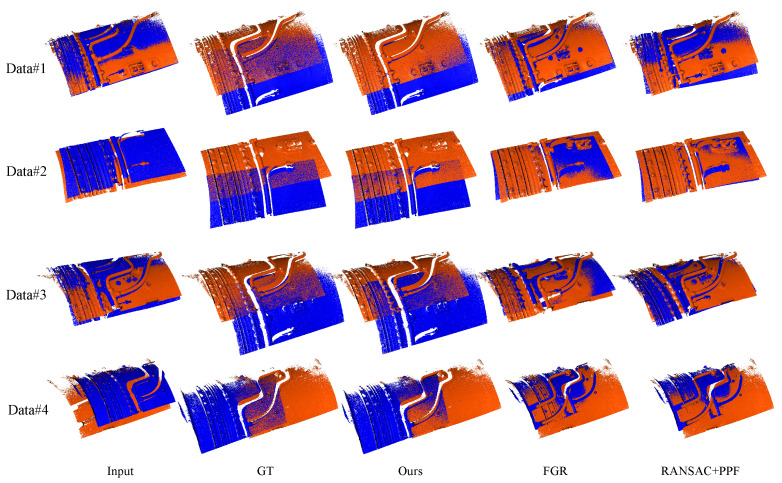
Pairwise registration results of the gas ducts, where the blue points represent the target point cloud and the orange points are from the source point cloud. It can be observed that RANSAC + PPF and FGR fail to achieve accurate registration. In contrast, PL-PPF shows satisfactory results that are closest to the real situation.

**Table 1 sensors-24-03358-t001:** Quantitative results of pairwise registration for the gas pipeline. The RMSE (in millimeters) and computation time (in seconds) for each pair of point clouds in Figure 7 are presented in this table. Num represents the total number of points in the input point clouds. Our method achieves the lowest RMSE values among all experiments, and the computation time is comparable to FGR but significantly better than RANSAC + PPF.

Index	Num(×10^4^)	FGR		RANSAC + PPF		Ours	
		RMSE	Time	RMSE	Time	RMSE	Time
Data#1	294	20.186	1.943	18.675	52.732	0.728	4.278
Data#2	278	16.722	1.615	17.453	43.874	0.106	4.322
Data#3	282	43.665	1.334	48.453	48.345	1.220	5.160
Data#4	297	51.789	1.673	43.538	49.650	0.403	4.856
Average	-	33.0905	1.64125	32.02975	48.65025	0.61425	4.654

We also provide a quantitative comparison of each paired registration method. To measure the accuracy of the registration, we use the root mean square error (RMSE) between the transformed point cloud $P=\left\{p_{i}\right\}$ and the ground truth $\hat{P}=\left\{{\hat{p}}_{i}\right\}$, where both point clouds represent identical structures at different spatial positions. The formula for calculating the RMSE is as follows:(8)$$RMSE=\sqrt{\frac{1}{n}\sum_{i=1}^{n}{\left(p_{i}-{\hat{p}}_{i}\right)}^{2}}$$
where pi and ${\hat{p}}_{i}$ are corresponding points in *P* and $\hat{P}$, respectively. RMSE directly relates to the accuracy of registration. A lower RMSE value indicates higher accuracy in paired registration. As shown in Table 1, our method achieves the lowest RMSE value, while RANSAC + PPF and FGR exhibit significantly higher RMSE values compared to our method. The higher RMSE values in these methods are primarily attributed to mismatches in their registration results. In terms of computational time, our method slightly surpasses FGR but significantly outperforms RANSAC + PPF. This is because our registration process does not involve iterative searching for correspondences, similar to FGR, where the iterative process has been found to be the most time-consuming step in global registration [6]. Consequently, our algorithm demonstrates superior performance among the three methods, delivering competitive computational efficiency. In summary, the results of the comparative analysis indicate that our proposed algorithm outperforms other registration methods in terms of accuracy, with a competitive computational speed.

### 5.2. Oil Ducts of Aircraft Engines

The oil system pipeline in an aircraft engine is a complex network responsible for delivering lubricating oil to critical engine components, ensuring their proper functioning and reducing wear and friction. This system consists of various types of pipes, such as oil delivery pipes, oil return pipes, lubricating oil cooling pipes, and lubricating oil supply pipes. Compared to the air system pipeline, the oil system pipeline has smaller diameters and less prominent features due to the lower fluidity requirements of lubricating oil. However, its layout is relatively more complex, with numerous critical attachments, demanding higher accuracy in registration.

Figure 8 displays the results of two sets of aircraft engine oil pipeline data under different registration methods. Our algorithm demonstrates superior performance even in more complex pipeline systems. In data1 and data2, both FGR and RANSAC + PPF fail to identify the features accurately, leading to incorrect registration. In data3, FGR’s registration result is incorrect, while our method still achieves higher accuracy compared to RANSAC + PPF. Data4 and data5 are oil pipelines with many obstructions. Our method can achieve correct registration. In contrast, the registration results of other methods are unsatisfactory. Notably, our method successfully matches the point clouds of the original scanned aircraft engine oil pipeline system.

**Figure 8 sensors-24-03358-f008:**
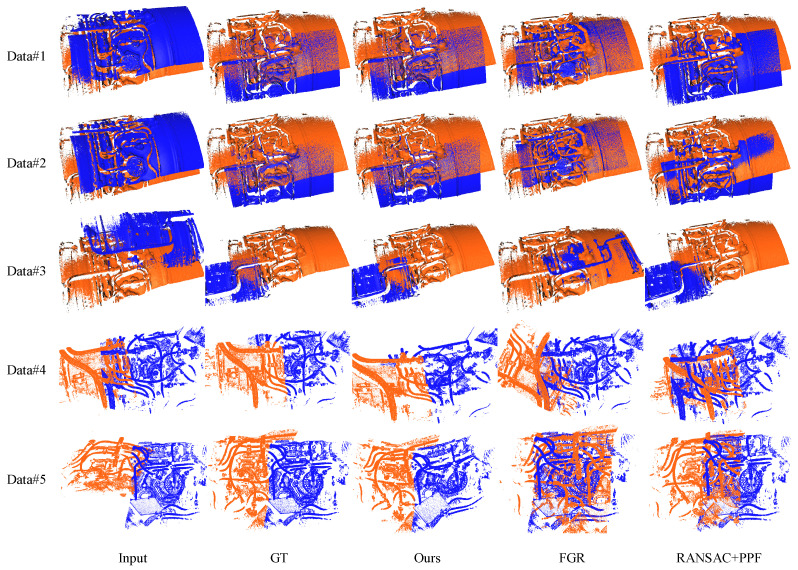
Pairwise registration results of the oil ducts, where the blue represents the target point cloud and the orange represents the source point cloud. Similar to the gas ducts, both FGR and RANSAC + PPF failed to achieve accurate registration, while PL-PPF demonstrates excellent performance in duct system registration.

The average values in Table 2 further confirm the effectiveness of our method in registering more complex aircraft engine oil pipeline systems. Our algorithm outperforms the other two methods and exhibits competitive computation speed.

**Table 2 sensors-24-03358-t002:** Quantitative results of pairwise registration for the oil pipeline. The RMSE (in millimeters) and computation time (in seconds) for each pair of point clouds in Figure 8 are presented in this table. Num is defined as in Table 1. Our method achieves high accuracy while maintaining competitive speed.

Index	Num(×10^4^)	FGR		RANSAC + PPF		Ours	
		RMSE	Time	RMSE	Time	RMSE	Time
Data#1	244	18.342	1.883	12.443	45.453	0.458	6.082
Data#2	264	21.554	1.237	18.346	51.563	0.369	3.550
Data#3	272	25.452	1.209	34.665	44.550	1.997	4.129
Data#4	152	22.348	0.885	15.792	37.953	0.815	3.256
Data#5	166	19.336	0.901	35.679	40.226	0.574	3.674
Average	-	21.406	1.223	23.385	43.887	0.843	4.138

## 6. Conclusions

This paper introduces a comprehensive and innovative automatic registration framework specifically designed for the point clouds of aircraft engine pipeline systems. Within the ambit of this framework, a novel feature descriptor, termed PL-PPFs (point line–point pair features), is proposed. This descriptor is meticulously crafted, taking into account both points and lines that are extracted from the point clouds of aircraft engine pipelines. The primary motivation behind the development of PL-PPFs is to significantly enhance the registration accuracy by effectively reducing mismatches, especially in scenarios characterized by incomplete or missing data. The inclusion of line features alongside point features in PL-PPFs is instrumental in providing a more robust description of the geometry and topology of the pipeline systems, thereby facilitating a more accurate and reliable matching process.

The PL-PPF descriptor leverages the intrinsic geometric relationships between points and lines to encode the spatial arrangement in a way that is more resilient to data sparsity and noise. By achieving this, it offers a considerable improvement over traditional point-based descriptors, especially in the challenging context of aircraft engine pipeline systems where the structural complexity and presence of occlusions and missing data can severely impede the registration process.

To validate the efficacy and robustness of our proposed framework and the PL-PPF descriptor, we conducted extensive experiments on a diverse set of aircraft engine pipeline system point cloud data. These experiments were designed to not only assess the accuracy of pairwise registration results but also to evaluate the resilience of PL-PPFs in handling data with varying degrees of completeness and noise levels.

The results of these experiments unequivocally demonstrate that our method achieves superior registration outcomes, surpassing existing approaches in terms of accuracy and reliability. Furthermore, our method exhibited remarkable resilience in the face of incomplete data, consistently delivering satisfactory registration results. This underscores the potential of the PL-PPF descriptor and the corresponding registration framework to serve as a powerful tool for applications involving the reconstruction, analysis, and maintenance of aircraft engine pipeline systems. 

## Figures and Tables

**Figure 1 sensors-24-03358-f001:**
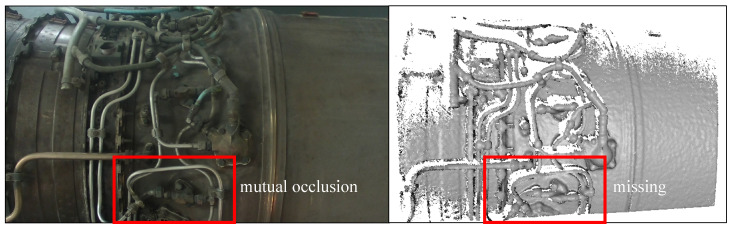
Image (**left**) and point cloud (**right**) of aircraft engine pipeline system. It can be seen that there are occlusions and data missing in the point cloud, which pose challenges for point cloud registration.

**Figure 2 sensors-24-03358-f002:**
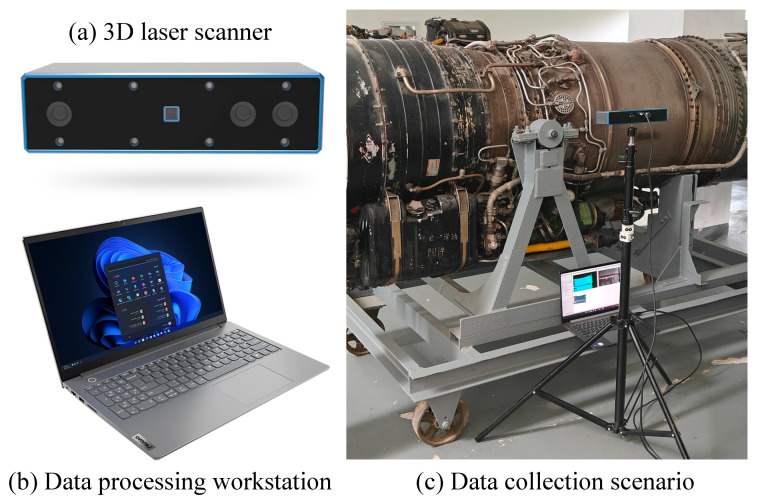
Hardware system for aircraft engine pipeline point cloud acquisition: (**a**) structured light 3D camera; (**b**) workstation; (**c**) acquisition scene. The 3D scanner is mounted on a tripod.

**Figure 3 sensors-24-03358-f003:**
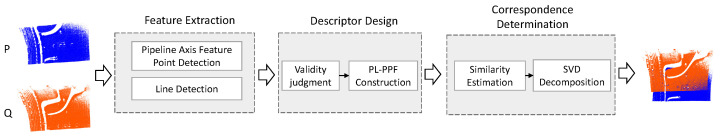
Overview of our proposed method. The presented approach entails the utilization of two point clouds, P and Q, as inputs. Within the feature extraction module, axis feature detection and line detection methodologies are employed to extract geometric features from the point clouds. Subsequently, the extracted features undergo validation, and PL-PPF descriptors are constructed. Finally, these PL-PPF descriptors derived from both point clouds are employed to evaluate their similarity and undergo singular value decomposition (SVD) solving.

**Figure 4 sensors-24-03358-f004:**
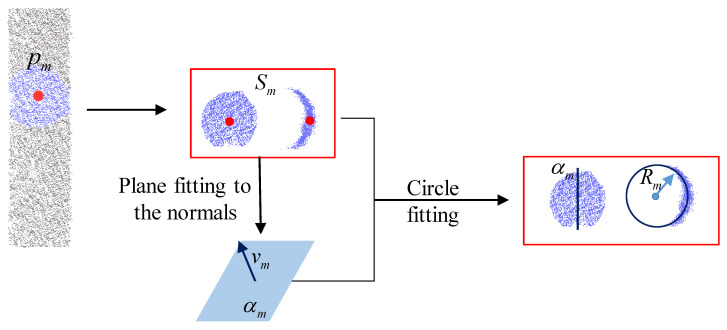
Illustration of pipeline feature detection. First, a point $p_{m}$ is randomly selected from the 3D point cloud. Next, we extract a patch $S_{m}$ from the point cloud *P*, and the normals of the points within this patch are projected onto a Gaussian sphere, resulting in a normal vector $v_{m}$. $v_{m}$ and $p_{m}$ can determine a plane. Then, $S_{m}$ is projected onto this plane, and a circle is fitted with $O_{m}$ as the center and $R_{m}$ as the radius.

**Figure 5 sensors-24-03358-f005:**
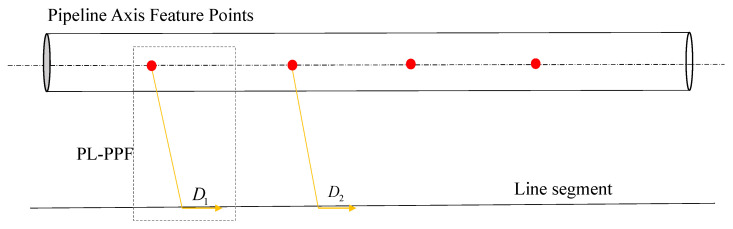
Illustration of ambiguity when combining feature points and line segment. The red points represent the axial feature points extracted from the pipeline. $D_{1}$ and $D_{2}$ are two instances of the PL-PPF descriptor, both constructed from the same point cloud data. However, they exhibit similar structures, which can potentially lead to ambiguity in feature matching during the point cloud registration process.

**Figure 6 sensors-24-03358-f006:**
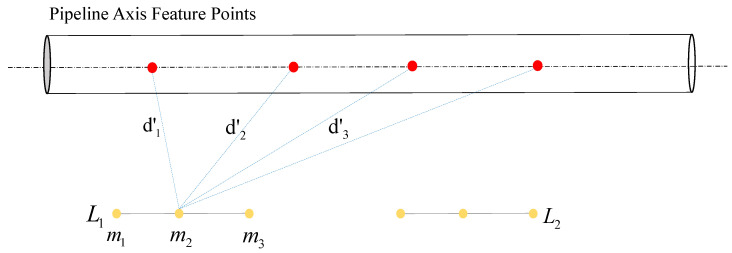
Data validity assessment. $L_{1}$ and $L_{2}$ are two line segments at different positions, where $m_{1}$ and $m_{3}$ are the endpoints of $L_{1}$, and $m_{2}$ is the midpoint of $L_{1}$. $d{\;}_{1}^{\prime }$, $d{\;}_{2}^{\prime }$, and $d{\;}_{3}^{\prime }$ represent the distances from the midpoint $m_{2}$ to the rivets. We can use PLScore to obtain the score for each line segment with respect to the pipeline axis feature points, and thus determine the most suitable pairing for descriptor construction.

## Data Availability

Data are contained within the article.

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
