# Peer review of "Robust Point Cloud Registration for Aircraft Engine Pipeline Systems"

_sensors, 2024, doi:10.3390/s24113358_

Round 1

Reviewer 1 Report (Previous Reviewer 1)

Comments and Suggestions for Authors

Dear authors

Thank you very much for addressing the comments and suggestions that this reviewer made to version 1 of the manuscript. I send my congratulations to the authors for their work.

In my opinion, version two of the manuscript sensors-2933112 is very clear and will allow readers to better understand the work.

I trust that this article will be published in the journal and will be well received among the scientific community.

Best regards.

Author Response

Thank you very much for taking the time to review this manuscript.  Thank you for your valuable advice!

Reviewer 2 Report (Previous Reviewer 2)

Comments and Suggestions for Authors

As I already evaluated in the previous revision, I state again that the article is interesting and brings new knowledge. Some typos and errors have been removed, but I couldn't find answers to some questions:

1) What is checked by 3D scanning on aircraft engines (not clear enough from the article). Is it an inspection of dimensional accuracy, pipe position or perhaps the completeness of individual components? In my opinion, it is primarily necessary to ensure the tightness of the pipeline, but 3D scanning is not suitable for this.

2) The article describes the methods of registration and its reliability. However, a practical example of the use of this method in practice is missing. Is it even realistic?

3) Would it not be possible to use 3D scanning systems that use a tracking system to determine the absolute coordinate system of individual images? The author states that it is unsuitable due to only small openings. And for another scanner, limited access is not a problem?

If these questions are answered and the reviewer did not find passages that explain the questions, please highlight or link to those parts of the article.

After completing or explaining the given questions, I recommend for publication.

Author Response

This manuscript is a resubmission of an earlier submission. The following is a list of the peer review reports and author responses from that submission.

Round 1

Reviewer 1 Report

Comments and Suggestions for Authors

Review Report Form sensors-2581703

Journal: Sensors (ISSN 1424-8220)

Manuscript ID: sensors-2581703

Title: Robust Point Cloud Registration for Aircraft Engine Pipeline Systems

General comments

The article focuses on a topic of great interest and tries to solve a problem related to aircraft engine inspection work. The article proposes a methodology to make 3D models of the aircraft engine pipeline systems in order to carry out inspections and identify possible structural or/and functional failures.

The methodological proposal is not completely new since the authors propose an improvement of an existing methodology (proposed in DOI: 10.1109/CVPR.2010.5540108 https://ieeexplore.ieee.org/document/5540108) creating a modification (new 3D descriptor called PL-PPF (Point Line-Point Pair Features)”)

Structure

Point S1. The text of the article is too descriptive: it dedicates a lot of space to the “Introduction” and “Related work” sections, however it does not dedicate much space to the description of the material used in the field tests (hardware system) neither Results and Conclusions.

Point S2. Within the “Method” section there is a section dedicated to Motivation: this section is repetitive since it has already been addressed in the “Introduction” section.

Point S3. Some ideas are repeated throughout the entire text. For example, the idea explained in the third paragraph of page 8 ("In order to...pipeline axis.") has already been described in the Introduction. It is convenient not to repeat concepts and ideas so that the manuscript is easier to read.

Contents

Point C1. The first thing that strikes me about the text is that it does not show data on the affiliation of the authors: this must be solved unless the reason is to preserve the anonymity of the authors.

Point C2. Another aspect that makes revision difficult is that the lines of the text are not numbered: if the lines were numbered it would be easier to identify the references to the reviewers' text.

Point C3. In addition to the Results shown (RMSEs and computation time of Tables 1 and 2) it would be interesting to add some more Results that allow verifying the effectiveness of the method to avoid failures due to occlusions, compared to the other methods explored in the work.

Point C4. The Conclusions of the work can be improved. One possibility is to list the conclusions related to each objective of the work (points cited in pages 2-end and 3-first).

Misprints or mistakes in important descriptions

Point M2. First paragraph of page 8: Now the text is “The Point Pair Feature (PPF), originally proposed by Bertram et al. [9], is a well-established…” but the reference [9], is “Drost, B.; Ulrich, M.; Navab, N.; Ilic, S. Model globally, match locally: Efficient and robust 3D object recognition. In Proceedings of the 2010 IEEE computer society conference on computer vision and pattern recognition. Ieee, 2010, pp. 998–1005.”. The authors must carefully review all the bibliographical references of the manuscript.

Point M2. Last paragraph of page 11: “We conducted tests on three sets of aircraft engine gas ducting system data and four sets of aircraft engine oil ducts to evaluate the performance of our method. The registration results for the gas ducts are presented in Figure 7, while the results for the oil ducts are shown in Figure 8.”… However, in Figure 7 (gas ducts) 4 datasets are shown (instead of 3) and in figure 8 (oil ducts) 3 datasets are shown (instead of 4). Authors must correctly identify the number of datasets used for gas and oil pipelines (as well as the results obtained). This must be clarified by the authors in the new manuscript.

Point M3. Last paragraph of page 12: “Table 1 [CHH: Plase use cite commond to cite figures and tables]”; I understand it is a mistake…

Figures

In general, the figures provide little information or are imprecise:

Point F1. Figure 1: what is the photograph (I suppose it is on the left) and what is the cloud of points (I suppose it is on the right)? Both are not identified. An area is identified as "mutual occlusion”, but more similar areas are seen that can also cause "missing zones": why is that area of the image marked and not anotherone?

Figure 2: shows a camera image (“3d laser scanner” is the same that “Structured Light 3D Camera”?), a computer but the information of interest (“the acquisition scene”) is not clearly shown.

Point F2. Figure 3: refers to 3 stages “A) Feature Extraction; B) Descriptor Design and C) Correspondence Determination”; each stage contains different processes (e.g. B) Descriptor Design : B1) Validity judgment and B2) PL-PPF Construction ) but in the descriptive sections of each stage other processes are described (e.g. B) Descriptor Design in 3.4 pages 8-9: Traditional PPF; line segmentation; Validity judgment and PL-PPF design). The non-correspondence between the Figures and the text descriptions make it difficult to correctly interpret the manuscript.

Point F3. Figure 4. The caption of Figure 4 is “Illustration of pipeline feature detection. First, a point pm is randomly selected from the pipeline. Next,…” It is correct or the correct sentence may be “Illustration of pipeline feature detection. First, a point pm is randomly selected from the 3Dpoint cloud. Next,…”

Point F4. Figure 5 and caption. Is you the same concept “feature line” and “Line segment”? now I interpret that they are different concepts.

Point F5. Figures 7 and 8: Authors have described in the caption what is “ GT” and “FGR”. On the other hand, it would be preferable to put “PL-PPF” instead of “ours”

Tables

Point T1. The nomenclature of the tables (FGR, PPF and Ours) is different from that used in the figures (Input, GT, Ours, FGR and RANSAC+PPF). This must be reviewed by the authors so that there is a perfect correspondence between what is shown in the figures and the data in the tables.

Author Response

Please find the responses in the attachment

Reviewer 2 Report

Comments and Suggestions for Authors

The article is quite interesting and brings new insights. After completing and correcting minor errors, I recommend publishing. 

Please add / explain better if necessary:

The proposed registration method is undoubtedly interesting. However, the reviewer is not sure if 3D scanning is suitable and reliable enough for the inspection of aircraft engines.

Alternatively, explain what is inspected by 3D scanning on aircraft engines (not sufficiently clear from the article). Is it an inspection of dimensional accuracy, pipe position or perhaps the completeness of individual components? In my opinion, the main thing to ensure is the tightness of the piping, but 3D scanning is unsuitable for this. 

In the article, the registration methods and their reliability are given. However, a practical example of using this method in practice is completely missing. Is it even realistic?

Would it not be possible to use 3D scanning systems that use a tracking system to determine the absolute coordinate system of each image? The author states that this is inappropriate because of the small apertures only. And for another scanner, isn't the limited access a problem?

Formal flaws and errors:

0. Introduction - I recommend numbering the chapters starting from "1".

Chapter 3.1 is followed by chapter 3.3 (missing chapter 3.2)

Author Response

(The authors gave the same response as above.)

Reviewer 3 Report

Comments and Suggestions for Authors

The paper presented an automatic registration framework for aircraft engine pipeline systems point clouds using Point Line- Point Pair Features 3D descriptor. The paper is well written, and the method is explained very well.

However, two comments need to be addressed:

1 - The authors mentioned that there are end-to-end deep learning approaches in the literature for pint cloud registration. The authors should have justified using the old and traditional approach instead of the recent and most successful deep learning frameworks.

2- Moreover, the authors should have compared their proposed framework to these methods in their experiments to validate their methods against recent successful DL approaches in the literature. 

Author Response

(The authors gave the same response as above.)
